# Performance of a Pilot-Scale Continuous Flow Ozone-Based Hospital Wastewater Treatment System

**DOI:** 10.3390/antibiotics12050932

**Published:** 2023-05-19

**Authors:** Takashi Azuma, Miwa Katagiri, Naobumi Sasaki, Makoto Kuroda, Manabu Watanabe

**Affiliations:** 1Department of Pharmaceutical Sciences, Osaka Medical and Pharmaceutical University, Takatsuki 569-1094, Japan; takashi.azuma@ompu.ac.jp; 2Department of Surgery, Toho University Ohashi Medical Center, Tokyo 153-8515, Japan; 3Pathogen Genomics Center, National Institute of Infectious Diseases, Tokyo 162-8640, Japan; nsasaki@niid.go.jp

**Keywords:** antimicrobial resistance (AMR), hospital wastewater, continuous ozonation system, pilot study, metagenomics, antimicrobials resistant bacteria (ARB), antimicrobial resistance

## Abstract

Antimicrobial resistance (AMR) is becoming a global concern. Recently, research has emerged to evaluate the human and environmental health implications of wastewater from medical facilities and to identify acceptable wastewater treatment methods. In this study, a disinfection wastewater treatment system using an ozone-based continuous flow system was installed in a general hospital located in Japan. The effectiveness of antimicrobial-resistant bacteria (ARB) and antimicrobials in mitigating the environmental impact of hospital wastewater was evaluated. Metagenomic analysis was conducted to characterize the microorganisms in the wastewater before and after treatment. The results demonstrated that ozone treatment enables effective inactivation of general gut bacteria, including *Bacteroides*, *Prevotella*, *Escherichia coli*, *Klebsiella*, DNA molecules, and ARGs, as well as antimicrobials. Azithromycin and doxycycline removal rates were >99% immediately after treatment, and levofloxacin and vancomycin removal rates remained between 90% and 97% for approximately one month. Clarithromycin was more readily removed than the other antimicrobials (81–91%), and no clear removal trend was observed for ampicillin. Our findings provide a better understanding of the environmental management of hospital wastewater and enhance the effectiveness of disinfection wastewater treatment systems at medical facilities for mitigating the discharge of pollutants into aquatic environments.

## 1. Introduction

In recent years, antimicrobial resistance (AMR) has become dangerously close to our daily lives, raising concerns about sustainable human development [1,2,3]. A significant issue with AMR is that it not only poses a direct health risk of infecting people through hospital- and community-acquired infections, but also an indirect risk of infecting humans via the environment [4,5]. O’Neill Commission, under the request of the UK government, estimated that if effective measures are not taken to combat the prevalence of AMR, annual global deaths from AMR will increase from 0.7 million in 2014 to 10 million by 2050, surpassing cancer-caused deaths, whereas the economic loss to global GDP will be $100 trillion [6]. According to the latest report in 2022 published in *The Lancet*, an internationally renowned medical journal, annual deaths attributable to AMR are anticipated to nearly double in five years to 1.27 million in 2019 [7]. The World Health Organization (WHO) has proposed the “One Health” approach as a comprehensive measure for humans-animals-environment and has called for the formulation of national action plans in each country [8]. In Japan, action plans have been established and measures are progressing [2,9].

Previous research has indicated the impact of AMR originating from wastewater entering the environment, despite the fact that the origins of AMR detected in aquatic environments are diverse [3,10,11]. Social interest in the risk management of wastewater originating from medical facilities, such as hospitals, has been rapidly growing [12,13,14]. Previous studies have reported antimicrobial-resistant bacteria (ARB) as a problem in clinical practice, suggesting that hospital wastewater may serve as a reservoir for AMR [15,16,17,18,19]. In addition to ARB, residual antimicrobials can be detected in hospital effluents [12,20,21]. The pollution load of hospital wastewater in aquatic environments ranges from several tens to 71%, with rates varying by country [22,23,24,25,26].

Human environmental risks from AMR environmental contamination include the health impacts associated with direct inoculation exposure to viable bacteria surviving in the environment [27,28,29]. Furthermore, it is important to consider the indirect effects of the propagation of antimicrobial resistance genes (ARGs) on *E. coli*, which are widely distributed in the environment, by encouraging the development of new AMR, even though the direct health effects of ARGs are considered minor, as is the case with viable bacteria [30,31,32]. Moreover, antimicrobials detected in the aquatic environment could be a contributing factor to the unexpected emergence of AMR from the environment, in addition to the toxic effects on the river environment [33,34,35]. Under these circumstances, assessing the actual situation of ARB and antimicrobials in hospital wastewater and their environmental risk, as well as seeking initiatives to develop treatments that can reduce or eliminate such risks, would contribute to protecting human health as well as improving the quality of medical care [36,37,38]. Furthermore, ARB is considered an important issue in ensuring the safety of the water environment and watershed preservation to secure healthy drinking and irrigation water [39,40].

With the recent remarkable development of science and technology, wastewater treatment systems that are effective in treating hospital wastewater have been developed [41,42,43,44,45]. Among them, ozone treatment has been the focus of research in recent years because it has strong sterilization and pollutant removal potential, including chemical-free deodorizing and residue-free wastewater after treatment [46,47,48]. However, the efficacy of wastewater systems based on ozone treatment for hospital wastewater has been primarily evaluated in small-scale (several hundred milliliters to several L) test systems in laboratories [49,50], with minimal research done on the actual hospital wastewater scale [51,52]. Our research group installed an ozone-based hospital wastewater treatment system in a hospital facility located in an urban Japanese region. A medium-scale batch-type treatment tank (effective volume of 1 m^3^) was installed in the hospital to evaluate the inactivation effect of ozone treatment on ARB and antimicrobials. Both ARB and residual antimicrobials were reduced to ≤1% after 20 min of treatment [21]. If the wastewater treatment of a hospital on a practical scale is proven to be a feasible solution to environmental AMR concerns, it may be conceived as an effective mechanism against AMR, thus contributing to the One Health initiative [32,53,54]. Furthermore, the results can potentially contribute to both the public interest in regional security and the safety of the local population [55,56,57].

As part of the efforts to implement a hospital wastewater treatment system in society, a pioneering trial has begun to verify the effectiveness of measures utilized to reduce the environmental burden of the hospital wastewater treatment system by conducting continuous treatment of the entire hospital wastewater before discharging it into the public sewer system. Therefore, in the present study, a continuous treatment system that can effectively treat hospital wastewater without interfering with hospital operations was developed. To achieve more effective treatment, we additionally tested an ultraviolet light-emitting diode (UV-LED) [58,59], which has recently been shown to be effective in disinfecting pathogenic microorganisms, including SARS-CoV2 [60,61,62], and is rapidly becoming more widely used. The disinfection process, which uses ozone plus an ozone catalyst, has previously been reported for wastewater treatment plants [63,64,65]. In cases such as hospital wastewater where it is virtually impossible to install large-scale treatment facilities or secure a new hospital site for wastewater treatment, the evaluation of the effectiveness of direct disinfection treatment for untreated raw hospital wastewater is a new challenge of great social interest in terms of hospitals and the environment [66,67]. The effect of applying disinfection wastewater treatment systems to hospital wastewater to reduce the burden of AMR in the environment was determined by clarifying the inactivation effect on ARB, ARGs, and antimicrobials, and analyzing the characteristics of microorganisms in wastewater before and after treatment using metagenomic analysis.

## 2. Materials and Methods

### 2.1. Hospital Wastewater Treatment Using Ozone Treatment Based on the Continuous Flow System

An ozone inactivation system for bacteria and antimicrobials present in hospital wastewater based on a continuous flow system was installed at the University Hospital, Ohashi Medical Center (BN; 35.652578° N, 139.683959° E), with a capacity of 319 beds, in Toho University, Tokyo, Japan, as previously reported [21]. Various wastewaters generated as a result of hospital activities were stored in two underground wastewater tanks (influent) with a total volume of 22.5 m^3^. The annual daily inflow of the wastewater tanks was 50 m^3^/day, which was approximately equal to the volume of the wastewater tanks. The supernatant was pumped directly into the public sewage system at 2 m^3^ per discharge with an average frequency of 25 discharges per day (50 m^3^/day as outflow). First, hospital wastewater from one of the two storage tanks was introduced into wastewater treatment tank 1 (ozone) at a flow rate of 20 L/min, with an effective volume of 1 m^3^ for ozone treatment. The ozone-treated wastewater was then flowed into wastewater treatment tank 2 (UV-LED) with an effective volume of 1 m^3^, which was connected to a UV disinfection unit for further inactivation owing to overflow inflow at a flow rate of 20 L/min. Finally, the treated wastewater was returned to the original storage tank on one side using a return pump (flow rate 150 L/min) for circulation to the original storage tanks (influent). The appearance and configuration of the ozone treatment system used in this study are shown in Figure 1, and a three-dimensional (3D) view of the equipment is shown in Appendix A.

Ozone was generated using an ozone generator (Ozonia^®^ TOGC45X, Suez Environment, Paris, France) equipped with a Pressure Swing Adsorption (PSA) oxygen generator. The hospital wastewater in the wastewater treatment tank was circulated using a circulation pump (32LPS5.75E, Ebara Corporation, Tokyo, Japan) at a flow rate of 5 L/min, and ozone gas was fine-bubbled through fine-bubble generating nozzles (YJ-9, For EARTH Co., Ltd., Tokyo, Japan) (microbubbles with diameters between 1 μm and 100 μm, using mode 30–50 μm, and ultrafine bubbles with diameters less than 1 μm in the 50–200 nm [68,69]) and introduced into the wastewater treatment tank. The ozone treatment was performed at an ozone generation rate of 27.5 g/h. UV irradiation was performed using a UV-LED system (DWM13-S06-XX-K, NIKKISO Co., Ltd., Tokyo, Japan) with a peak emission of 280 nm and an intensity of 43 mW/cm^2^. A portion (1 L) of the solution in this tank was collected 0, 1, 4, 6, 8, 15, and 29 days after the start of the experiment. Basic water quality parameters (biochemical oxygen demand (BOD), chemical oxygen demand (COD), suspended solids (SS), and total nitrogen (TN)) during treatment in this investigation, along with DNA concentration and total reads of metagenomic DNA sequences, are shown in Appendix A. Sodium thiosulfate was immediately added to mitigate the effects of residual ozone on the samples [70,71]. Samples were stored at 4 °C in the dark and processed for 12 h.

### 2.2. Viable Bacterial Counting of Wastewater Samples

To determine the efficacy of ozone treatment in inactivating potential *β*-lactam-resistant bacteria, an aliquot (100 µL) of influent or treated wastewater sample was 10-fold serially diluted with phosphate-buffered saline, followed by spreading on non-selective media BTB (Bromothymol Blue, lactose agar; Drigalski Agar, Modified) agar and CHROMagar ESBL plates (bioMérieux S.A., Marcy-l’Étoile, France) for extended-spectrum *β*-lactamases (ESBL) producing bacteria. Colony-forming units per mL (CFU/mL) were determined at the appropriate dilution for each treatment time point.

### 2.3. Metagenomic DNA-Seq Analysis of Wastewater Samples

To collect organisms larger than bacteria, ozone-treated wastewater samples were passed through TPP Rapid Filtermax Vacuum Filtration systems (TPP, Trasadingen, Switzerland) in 100 mL bottles fitted with 49 cm^2^, 0.2 µm polyethersulfone membranes. The membranes were removed from the bottles and stored at −30 °C until DNA extraction. One-fourth of the collected membrane, corresponding to 25 mL of influent or treated water, was cut into small pieces and placed in ZR-96 BashingBead Lysis Tubes (0.1 and 0.5 mm; Zymo Inc., Irvine, CA, USA). Bacterial lysis buffer (800 µL; Roche, Basel, Switzerland) was added to the bead tube, which was then frozen at −30 °C and thawed at 23 °C. The tube was subjected to bead-beating (1500 rpm for 10 min) using a GenoGrinder 2010 homogenizer. After brief centrifugation (8000× *g* for 3 min), 400 µL of the supernatant was collected. The DNA in the supernatant was purified using a Roche MagNa Pure Compact instrument (DNA_Bacteria_v3 protocol; elution:50 µL). DNA concentration and purity were measured using a Qubit DNA HS kit (Thermo Fisher Scientific, Waltham, MA, USA).

Metagenomic DNA-seq libraries were prepared using the QIAseq FX DNA Library Kit (Qiagen, Hilden, Germany), followed by short-read sequencing using the NextSeq 500 platform (2 × 150-mer paired-end) (Illumina, San Diego, CA, USA). Adapter and low-quality sequences were trimmed using Sickle version 1.33 (https://github.com/najoshi/sickle) considering the following parameters: average quality threshold “−q 20” and minimum length threshold “−l 40.” Metagenomic DNA-seq analysis was performed using clean reads for homology searches without de novo assembly for all subsequent analyses. Detailed scripts and databases are described below.

Taxonomic classification of every read from the metagenomic analysis was performed using mega-BLAST (e-value threshold, 1E^−20^; identity threshold, 95%) against the NCBI nt database using MePIC2 [72] and was subsequently analyzed using MEGAN 6 [73]. Statistical analysis by ozone and subsequent UV-LED treatments was analyzed using two-way repeated ANOVA (R-Studio 2022.12.0+353).

Resistome analysis using metagenomic DNA-seq reads was performed using ARGs-OAP v3.2.1 against the implemented ARG database [74,75].

All raw read sequence files are available from the DRA/SRA database (Appendix A).

### 2.4. Analytical Procedures for Antimicrobials

A total of six antimicrobials grouped into five classes, *β*-lactams (ampicillin), new quinolones (levofloxacin), macrolides (azithromycin and clarithromycin), tetracyclines (doxycycline), and glycopeptide (vancomycin) (>98%), were examined in the present investigation on the basis of a previous report on their concentrations and detection frequencies in hospital effluent, wastewater, and river water, both in Japan and around the world [25,76], as well as on the basis of antimicrobial use in clinical sites in Japan [77,78].

The concentrations of the target antimicrobials in the wastewater were analyzed as previously described [21,25]. Briefly, 10 mL of wastewater was filtered through a glass fiber filter (GF/B, 1 μm pore size, Whatman, Maidstone, UK), and the solutions were subjected to OASIS HLB solid-phase extraction cartridges (Waters Corp., Milford, MA, USA) at a flow rate of 1 mL/min. The adsorbed antimicrobials were eluted with 3 mL acetone and 3 mL methanol and then evaporated mildly to dryness under a gentle stream of N_2_ gas at 37 °C. The residue was solubilized in 200 μL of a 90:10 (*v*/*v*) mixture of 0.1% formic acid solution in methanol, and 10 μL of this solution was analyzed using an ultra-performance liquid chromatography–tandem mass spectrometry (UPLC) system coupled to a tandem quadrupole mass spectrometer (TQD, Waters Corp.). Quantification was performed by subtracting the blank data from the corresponding data yielded by the spiked sample solutions to account for matrix effects and losses during sample extraction [79,80]. The recovery rates of antimicrobials in the wastewater influent ranged from 77% to 108% (Appendix A), and the limits of detection (LODs) and limits of quantification (LOQs) were calculated as the concentrations at signal-to-noise ratios of 3 and 10, respectively [81,82].

## 3. Results

### 3.1. Proportion of Bacteria in Hospital Wastewater after Ozone Treatment

Hospital wastewater was treated with ozone followed by UV-LED irradiation in a continuous-flow pilot plant (Figure 1) for 0, 1, 4, 6, 8, 15, and 29 days after starting continuous treatment. The visible brown color of the wastewater disappeared on day 1, and continuous treatment kept the treated water clear until the end of the treatment on day 29 (Figure 2A). Furthermore, the general water quality parameters (BOD, COD, SS, and TN) did not decrease or differ after the treatment (Appendix A).

Although it did not exhibit significant inactivation of the general parameters apart from the visible color, the treated water collected from the ozone or UV-LED tanks showed >90% inactivation of viable bacteria on BTB agar after 1 d of treatment (Figure 2B,C), and it maintained the level of reduced viable bacteria until day 29 (Figure 2C). In addition, the original storage tank (influent) had a reduced CFU of viable bacteria because the original storage tank (influent) received treated water (20 L/min) after ozone/UV-LED treatment from the bypass route (Figure 1). Compared with the bacterial CFU on BTB agar, the potential ESBL-producing Enterobacteriaceae on CHROMagar ESBL demonstrated a rather low susceptibility to ozone/UV-LED treatment, but finally underwent >90% inactivation on day 29 (Figure 2D).

To characterize the actual susceptibility of bacterial genera to ozone, membrane-trapped bacteria in the treated sample (corresponding to the 25 mL water sample) were subjected to genomic DNA extraction (Appendix A), and metagenomic DNA-seq analysis was performed (summarized in Appendix A). Notable fecal bacteria, *Bacteroides* and *Prevotella*, showed 90% less detection following ozone treatment compared with the original storage on day 1 (Figure 3). In addition, the most ubiquitous ESBL producers, *Escherichia* and *Klebsiella*, showed >80% less detection from the original storage on day 1 following treatment (Figure 3). On the other hand, the environmental bacteria *Acinetobacter* and *Pseudomonas*, which include potential nosocomial pathogens, exhibited increasing CFU after ozone/UV-LED treatment; however, there were low amounts of inputs from the influent tank for these two genera (Figure 3). Further taxonomic classification at the species level using the MEGAN 6 software suggested that possible bacterial isolates, which could be genetically similar to *Acinetobacter* sp. WCHAc010034, and *Pseudomonas* spp. LTGT-11-2Z significantly increased growth over 100-fold from day 0 onwards (Appendix A). *Acinetobacter* sp. WCHAc010034 was isolated from hospital sewage in China (BioSample SAMN05356835) and *Pseudomonas* sp. LTGT-11-2Z was isolated from the roots of Alhagi sparsifolia Shap. in the Taklamakan Desert, China (BioSample: SAMN10219285), suggesting that neither were identified as clinical specimens; however, their pathogenicity remains to be investigated.

### 3.2. Resistome Analysis in Hospital Wastewater Subjected to Ozone Treatment

In addition to the bacterial taxonomic analysis, ARG resistome analysis was performed using metagenomic DNA-seq reads. The top 12 most abundant ARGs with a combined display of specific numerical values and a composite display of colored bars are shown in Figure 4 (all the results obtained for other ARGs are summarized in Appendix A). Class 1 integrons (*sul1* and *qacEdelta*), *β*-lactamase GES variants (*bla*_GES-15_, *bla*_GES--14_, and *bla*_GES--5_), aminoglycoside acetyl transferase (*aac(6′)-31*), and tetracycline resistance (*tet*(39) and *tet*(36)) were mainly detected in sewage samples. Most of these were significantly inactivated to less than 10% of the sequencing reads on day 1, consistent with the results of the CFU (Figure 2) and metagenomic analyses (Figure 3). No increase in the ARGs was observed during the experiment (Appendix A). Additional statistical analysis showed that the effect of ozone and subsequent UV treatment on the wastewater was significant (adjusted *p*-value < 0.05, two-way repeated ANOVA and pairwise *t*-test) among *Bacteroides*, *Prevotella*, *Pseudomona*s, *Bifidobacterium*, and *Ruminococcus* families, and partial effects on the other bacterial families were also suggested (Appendix A).

### 3.3. Removal of Antimicrobials by Ozone Treatment

All six targeted antimicrobials were detected in the hospital wastewater before treatment. The detected concentrations of the antimicrobials ranged from 746 ng/L to a maximum of 37.9 μg/L, and the order of the detected concentrations was different for each compound. The detected concentration of each antimicrobial at the start of treatment was 17.9 μg/L ± 17.6 μg/L for ampicillin, 13.6 μg/L ± 6.1 μg/L for levofloxacin, 1.8 μg/L ± 1.5 μg/L for azithromycin, 1.4 μg/L ± 136 ng/L for clarithromycin, 1.1 μg/L ± 95 ng/L for doxycycline, and 11.0 μg/L ± 0.8 μg/L for vancomycin. LC–MS/MS parameters and validations of each antimicrobial and the validation of the recovery rates of antimicrobials in the wastewater and the limits of detection (LODs) and limits of quantification (LOQs) are summarized in Appendix A. Residual antimicrobials detected in hospital wastewater are thought to originate from antimicrobials used to treat diseases in clinical settings [83]. These values were largely similar to those reported in a survey conducted in an urban hospital located in a different region of Japan (734 ng/L to a maximum of 13.4 µg/L) [84], and largely consistent with those previously reported in other countries [12,85,86]. The time course of the antimicrobial concentrations in hospital wastewater during treatment is summarized in Figure 5, and the detailed concentrations are shown in Appendix A.

Ozone treatment was effective in removing all the targeted antimicrobials. More than 99% of azithromycin and doxycycline was removed immediately after treatment and was not detected in the ozone treatment tank throughout the experiment. The removal rates for levofloxacin and vancomycin remained between 90% and 97% during treatment. Clarithromycin was more readily removed than the other antimicrobials (81% to 91%), and no clear removal trend was observed for ampicillin. In the wastewater treatment tank where UV-LED irradiation followed ozonation, ampicillin and levofloxacin additionally decreased from their levels in the ozone treatment tank by an average of 15% and 53%, respectively. However, no notable changes in the concentration were observed for the other compounds. Finally, the concentration of antimicrobials before discharge into the public sewage system decreased during continuous-type treatment. The average removal rates of antimicrobials in the present study were 71 ± 24% for levofloxacin, 82 ± 16% for azithromycin, 88 ± 10% for doxycycline, and 44 ± 42% for vancomycin. However, no clear removal trends for ampicillin and clarithromycin were observed during the experiment, which was largely consistent with the removal trend of antimicrobials in the ozone treatment tank.

## 4. Discussion

The detection of ARB and antimicrobials in hospital wastewater throughout the period of this treatment test suggests that a certain amount of these pollutants is released from every hospital into the public sewage system, and that it is vital to implement social countermeasures on a large scale. The original storage tank (influent) showed an increase/decrease in the detected concentrations of both ARB and antimicrobials, which was associated with the continuous inflow of untreated raw hospital wastewater into the original tank.

General gut bacteria, including *Bacteroides*, *Prevotella*, *E. coli*, and *Klebsiella,* were significantly inactivated by ozone treatment (Figure 3). However, environmental bacteria, such as *Acinetobacter* and *Pseudomonas,* exhibited notable persistence to ozone, as in a previous batch treatment trial [21]. Based on the AMR issue, the persistence of *Acinetobacter* and *Pseudomonas* could play a major role in AMR reservoirs, although ozone treatment might act efficiently. Such persistence remains to be characterized in future studies to achieve a sustainable treatment process under continuous hospital operation. This study demonstrated that ozone treatment enabled the effective inactivation of viable bacteria (Figure 3) and DNA molecules, including ARGs (Figure 4); however, the general water quality parameters did not differ (BOD, COD, SS, and TN) (Appendix A). This finding strongly suggests that such parameters may not be essential criteria for reducing hospital-associated AMR factors because ozone causes partial damage to viable organisms and DNA molecules, leading to dead bacteria and damaged DNA molecules that no longer function.

The characteristics of the components most likely to be removed (>90% removal after 10 min treatment) were consistent for the antimicrobials and for ampicillin and clarithromycin, which were not adequately removed by the sequential treatment in this study; the removal rate was 96–100% for these antimicrobials at 40 min after treatment. These data suggest that it is essential to upgrade the treatment system and conduct demonstration tests and evaluations at an actual plant scale when developing from a batch (small-scale) system to the actual treatment of hospital wastewater.

The results of the present investigation show that ozone treatment removes antimicrobials, and the treatment time required for removal differs for each compound, supporting the results reported in previous evaluations of ozonation of pharmaceuticals in environmental water [87,88]. Antimicrobials, which tended to remain in the treated water compared to other antimicrobials, such as clarithromycin and ampicillin, could possibly be removed by increasing the ozone injection volume and prolonged treatment (high ozone exposure volume) [89,90]. Additionally, it would be effective to combine ozone treatment with other treatments for environmental pollutants that are difficult to treat effectively with ozone treatment alone. The fact that the UV-LED treatment in this study further improved the removal rates of ampicillin and levofloxacin [91,92], which are known to be easily degraded by light irradiation in the UV region, will prove useful when examining the effectiveness of the treatment for a wide spectrum of environmental pollutants in wastewater. Some antimicrobials, including *β*-lactam antimicrobials such as ampicillin, are attenuated in water within a few hours [93,94]. Clarithromycin is highly persistent in the environment as it is not susceptible to attenuation in the aquatic environment through photolysis or biodegradation [95,96]. Levofloxacin is considered the primary antimicrobial agent that causes clinical problems with ARB [97]. Previous studies have reported that ozone and/or UV treatment reduces ecotoxicological effects to approximately 1/10–1/20,000 compared to untreated compounds [47,98,99,100]. On the other hand, some researchers pointed out that the toxicity increases approximately 2–100 fold in some cases [99,100]. It is known that differences in susceptibility to these toxic effects occur in different target species, and that under conditions where multiple compounds coexist, weakening or strengthening effects would occur compared to exposure to a single substance [101,102]. Whole effluent toxicity (WET), as recommended by the US Environmental Protection Agency (EPA), which evaluates the toxic effects of a target water body as a whole using species at different trophic levels, would be a comprehensive approach to address these issues [103,104,105]. In addition, the strong oxidizing action of ozone or hydroxyl radicals can decrease the formation of transformation products by providing sufficient processing time and by acting in combination with catalysts, such as UV and hydrogen peroxide [46,106,107]. The results demonstrated that when antimicrobials with an environmental impact were effectively removed before flowing into the environment and kept at a low level, they are notable as an effective measure to reduce the environmental impact caused by AMR.

It is possible to reduce or eliminate the inflow of ARB and antimicrobials as environmental pollutants into the aquatic environment through the advancement of wastewater treatment systems [3,65,108]. In addition, it is important to conduct further optimization and refinement tests to maximize the synergistic effect of ozone treatment with an ozone catalyst, which is often applied to secondary-treated water after biological treatment in the wastewater treatment area [44,47,109]. However, a practical issue persists in that the cost of these advanced treatments increases as treatment systems become sophisticated and/or multiple treatments are combined [13,110]. Research that considers the cost aspects of actual implementation should be conducted in the future. The operating costs of this disinfectant system were approximately US $300 for one month using the maximum electronic power of the system, which could be an affordable cost for AMR disinfectants. The legalization of the required reduction levels in conjunction with research on environmental risk assessment for AMR discharge into the aquatic environment, as well as the promotion of the development of new mitigation strategies for dealing with AMR from an environmental perspective, needs to be emphasized. In addition, it will be challenging to deepen social understanding and support hospital incentives. Further development and return to society in both academia and industry are required.

As studies have revealed the antimicrobial-resistant nature of hospital wastewater and the significant impact of the loads discharged into the environment, more attention is being paid to how hospital wastewater should be treated [12,56]. However, research on environmental management and mitigation control of ARB and antimicrobials in hospital wastewater is still limited worldwide, owing to the general difficulty in researching hospital wastewater [111,112,113]. To the best of our knowledge, this is the first report on the effectiveness of a pilot-scale continuous wastewater treatment system based on ozonation for the inactivation of ARB and antimicrobials in the entire effluent generated by a hospital.

## 5. Limitations

The limitations of this study are as follows: The first is the optimization of wastewater treatment systems. An ideal wastewater treatment system involves the continuous direct discharge of treated water into a public sewage system. However, there are still issues that have not been fully covered in the present investigation in terms of practical aspects, such as technology, funding, and the need to reconstruct the entire hospital facility to continue treating the entire hospital wastewater while maintaining a balance with the constant inflow of untreated wastewater [114,115]. Another issue is the technical restrictions in maintaining the effect of UV light from UV-LED for a long period because raw hospital wastewater contains multiple solid organic substances. Further improvements in these aspects of the processing equipment need to be examined in the future.

Second, the inactivation effects of ARB and antimicrobials were investigated. Neither ARB nor antimicrobials were completely inactivated during the trial of hospital wastewater treatment. Under these conditions, microorganisms remain viable in the treated hospital wastewater. It is essential to elucidate the potential of these microorganisms to form biofilms in treatment tanks [116] and to evaluate the pathogenicity and environmental impact of microorganisms that require more ozone than other microorganisms [66,117].

Finally, the treatment effectiveness for the basic general water quality parameters was noted. For the hospital wastewater treatment in this study, the treatment system was found to be capable of inactivating ARB and antimicrobials, which have become new environmental pollutants of concern in recent years, although some improvements, including water quality, have not yet been achieved. Improving the treatment to include these water quality items can be achieved by increasing the amount of ozone injected, but in some respects, this treatment strategy is not practical or energy-efficient and it would be more effective in combination with other treatments [118,119,120]. These issues can be improved via biodegradation, as it is known that ozone treatment alone makes it difficult to convert a persistent substance into a biodegradable substance [47,49]. Our results support the need for further conclusive research considering experimental, technical, and regional customs, bias, and unknown factors.

## 6. Conclusions

In the present study, an ozonation-based continuous-flow disinfection wastewater treatment system was implemented in a core hospital located in the center of Japan, and its effectiveness in mitigating the environmental impact of AMR associated with hospital wastewater was evaluated. The results showed that both ARB and antimicrobials that would have an impact on the environment were effectively removed and maintained at a low level during treatment, which would be an effective countermeasure to mitigate the environmental impact caused by AMR. These findings are significant for implementing feasible and effective countermeasures to address AMR in the environment. The overall results facilitate a comprehensive understanding of the AMR risk posed by hospital wastewater and provide insights for devising strategies to eliminate or mitigate the burden of ARB and flow of antimicrobials into aquatic environments. Our findings could help enhance the effectiveness of introducing wastewater treatment systems, not only in wastewater treatment plants, but also in medical facilities, to reduce the discharge of pollutants into rivers, thereby contributing to environmental and human health safety.

## Figures and Tables

**Figure 1 antibiotics-12-00932-f001:**
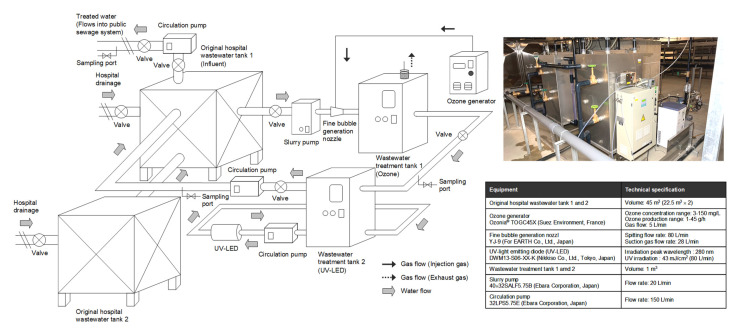
Schematic representation of pilot-scale continuous-flow ozone-based treatment system implemented in a hospital facility. The picture depicts the appearance of the hospital wastewater disinfection treatment system equipped with the ozone treatment system tested in this study. The technical specifications of the equipment used in the system are described in detail below. A 3D view of the equipment is shown in Appendix A.

**Figure 2 antibiotics-12-00932-f002:**
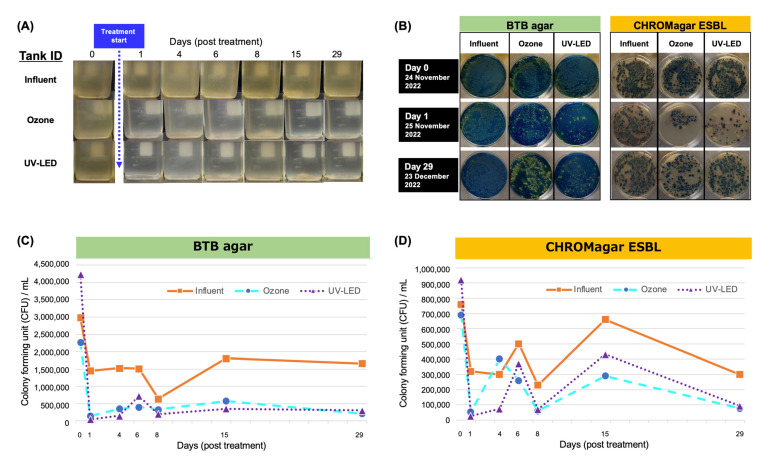
Characterization of treated wastewater by bacterial viability. (**A**) Visual image of the collected waste or treated wastewater after ozone or UV-LED treatment. (**B**) Isolation of bacteria from treated wastewater samples on BTB agar and CHROMagar ESBL. An aliquot (100 µL) of influent, ozone-, or UV-treated wastewater samples was spread on the agar plate at a 10-fold dilution. (**C**,**D**) Colony forming units per milliliter (CFU/mL) were determined at the appropriate dilution for each treatment time point.

**Figure 3 antibiotics-12-00932-f003:**
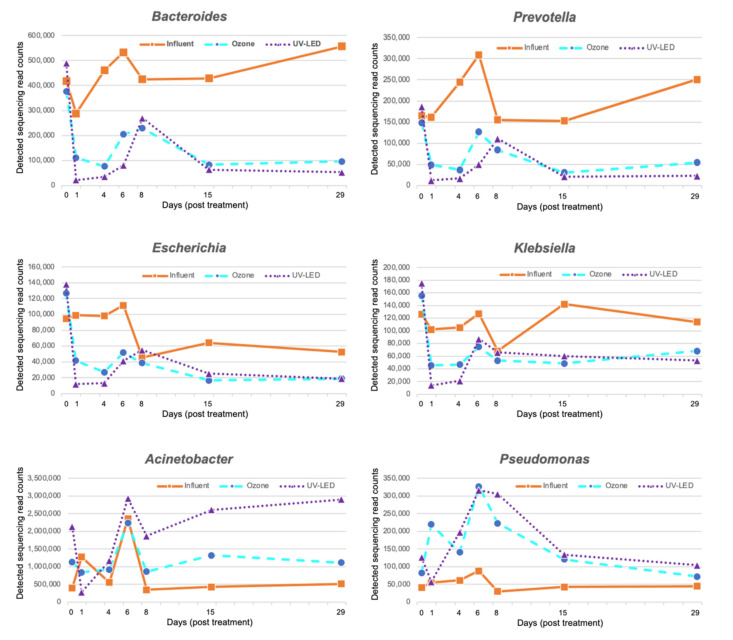
Metagenomic DNA-seq analysis of bacteria trapped on a 0.2 µm filter after ozone-UV-LED treatment. Notable bacterial genera (*Bacteroides*, *Prevotella*, *Escherichia*, *Klebsiella*, *Acinetobacter*, and *Pseudomonas*) were highlighted to show the metagenomic sequencing read counts detected by megablast search and subsequent taxonomic classification using the MEGAN 6 application. The results obtained for each bacterial genus are summarized in Appendix A.

**Figure 4 antibiotics-12-00932-f004:**
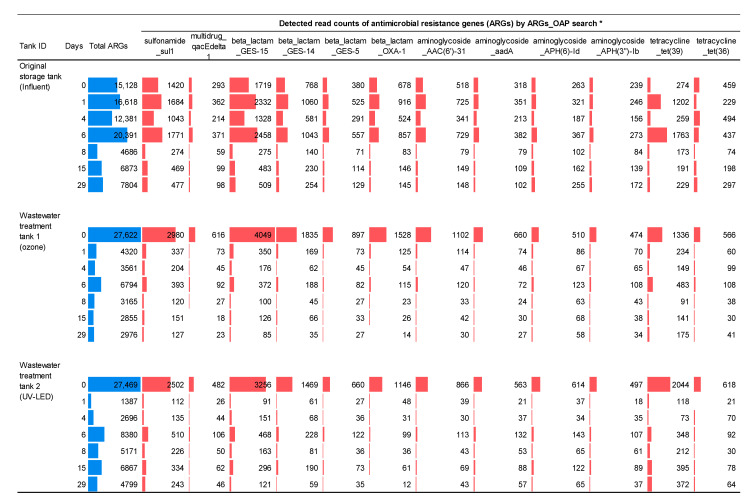
Metagenomic DNA-seq analysis of antimicrobial resistance genes (ARGs) after ozone-UV-LED treatment. Notable top 12 ARGs were selected to show the sequencing reads corresponding to the targeted ARGs using the ARGs_OAP program. DNA conc. (ng/µL) and Metagenomic DNA-seq (total reads) are shown in Appendix A, and all obtained results for other ARGs are summarized in Appendix A.

**Figure 5 antibiotics-12-00932-f005:**
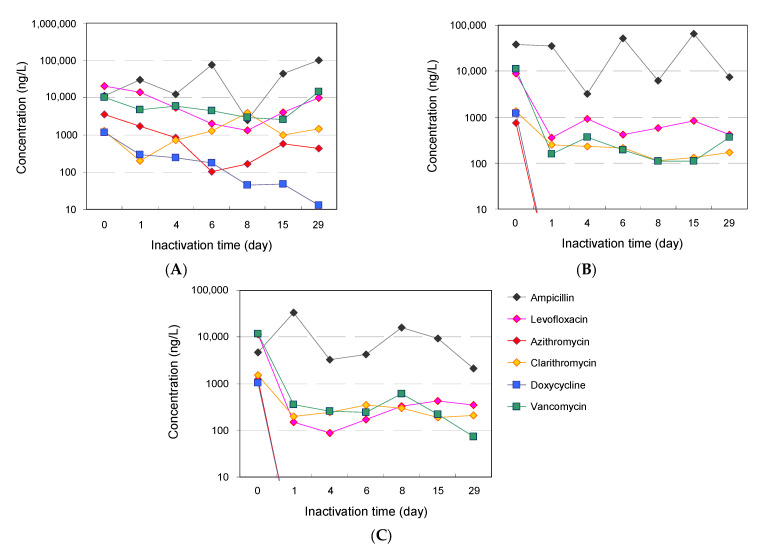
Time course of antimicrobial concentrations in hospital wastewater during ozone treatment. Removal of antimicrobials over time during treatment of hospital wastewater. (**A**): original storage tank (influent), (**B**): wastewater treatment tank 1 (ozone), and (**C**): wastewater treatment tank 2 (UV-LED). A summary of antimicrobial concentrations at each treatment time is shown in Appendix A.

## Data Availability

All raw read sequence files are available from the DRA/SRA database (accession numbers DRR439511–DRR439531 [see Appendix A).

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
