# Peer review of "Performance of a Pilot-Scale Continuous Flow Ozone-Based Hospital Wastewater Treatment System"

_antibiotics, 2023, doi:10.3390/antibiotics12050932_

Round 1

Reviewer 1 Report

Dear Editor,

The topic is very important and promised, however the performance is  very poor and not appropriate for publication in the current form.

A major revision id require:

1)    The treatment system is wrong. No primary and secondary treatment were conducted and indicated. The AOP (advanced oxidation process), which based on ozone or UV, always applied after secondary treatment. Therefore,  you cannot call the system that was applied in that manuscript -  an advanced treatment! You cannot apply AOP on raw WW and call it advanced treatment. The only treatment which was demonstrated in that manuscript – is a simple disinfection process that was done by a strong oxidant such as ozone and UV photons. For evidence, the oxidant was strong enough to disinfect bacteria but not strong enough to mineralize organics (BOD, COD, SS-  didn’t change during the treatment!!!). Additionally, the high concentrations of the BOD, COD  and SS, that characterize raw WW or insufficient treated effluent, may act as a scavengers in a real AOP treatment process (consumed immediately by the organics or and block (SS) the UV photons).

2)    The results claim to demonstrate that ozone treatment enables effective degradation of the antimicrobial molecules, however, the results demonstrate only a simple disinfection process rather than degradation.

The only degradation which obtained, is transformation of the parent compounds (PCs)  create only  transformation products (TPs) that are slightly differ (chemically) from the parent compounds, however, these TPs mostly more persistent, less biodegradable and could be even more toxic than the PCs. The amount of ozone which uses in that study was enough for disinfection only and slightly transformed the PCs) – I can’t indicate on an AOP process and mechanism

3)    No sufficient explanation regarding the AOP applications in the Introduction

4)    Please show UPLC gradient conditions, the column type? and additional analytical values for MS. what was the LOD? LOQ? validation process of the analytical method?

5)    No matrix effect was indicated due to the high amount of organics in the non-sufficient treated WW?

6)    Line 239 - It is wearied, how the water became clearer and the measured parameters didn't change?!!

7)    Figure 4 is a table not a figure.

 The limitations showed by the authors and the describe above require a major revision of the system, experiment, results and discussion prior to publication

Author Response

Reviewer1

Comments and Suggestions for Authors

Dear Editor,

The topic is very important and promised, however the performance is  very poor and not appropriate for publication in the current form.

A major revision id require:

1)    The treatment system is wrong. No primary and secondary treatment were conducted and indicated. The AOP (advanced oxidation process), which based on ozone or UV, always applied after secondary treatment. Therefore, you cannot call the system that was applied in that manuscript - an advanced treatment! You cannot apply AOP on raw WW and call it advanced treatment. The only treatment which was demonstrated in that manuscript – is a simple disinfection process that was done by a strong oxidant such as ozone and UV photons. For evidence, the oxidant was strong enough to disinfect bacteria but not strong enough to mineralize organics (BOD, COD, SS- didn’t change during the treatment!!!). Additionally, the high concentrations of the BOD, COD and SS, that characterize raw WW or insufficient treated effluent, may act as a scavengers in a real AOP treatment process (consumed immediately by the organics or and block (SS) the UV photons).

Response:

Thank you for the careful observation and invaluable contribution. Our research group understands the points raised by the reviewers about the processing method, and we discussed them before starting this project. First, regarding the advanced oxidation process (AOP), the hospital wastewater treatment investigation conducted in this study are totally different from those conducted at drinking water plants and wastewater treatment plants. The most unique feature of hospital wastewater treatment is that wastewater from existing hospital facilities is discharged directly into the public sewer system in Japan without treatment. This would mean that if a new hospital wastewater treatment system were to be installed, a large treatment facility would be required to occupy a certain amount of space in the hospital in the wastewater treatment area, or purchase a large new parcel of land next to the hospital and build a treatment facility. This trend is similar not only in Japan, but also in other parts of the world. Due to these practical limitations, it is difficult to install sedimentation tanks or biological treatment reactors, so this study is a new perspective and an unprecedented challenge to evaluate the effectiveness of measures that can be taken by hospital facilities that can directly treat raw sewage.

For this reason, the term "advanced treatment" is used in the text to refer to the broad term advanced wastewater treatment, which is also used for nitrogen and phosphorus removal in biological treatment before secondary treatment, rather than limiting the term directly to AOP as a specific form of treatment. There are parts of the content that are unclear and difficult to understand, so in accordance with the reviewers' opinions, we have revised the main text along with additional references to clarify that advanced treatment refers to advanced wastewater treatment using ozone treatment as follows:

“As part of the efforts to implement the hospital wastewater treatment system in society, a pioneering trial has begun to verify the effectiveness of measures utilized to reduce the environmental burden of hospital wastewater treatment system by conducting continuous treatment of the entire hospital wastewater before discharging it into the public sewer system. Therefore, in the present study, a continuous treatment system that can effectively treat hospital wastewater without any interference with hospital operations was developed. To achieve a more effective treatment, we additionally tested an ultraviolet light-emitting diode (UV-LED) [58,59], which has recently been shown to be effective in disinfecting pathogenic microorganisms, including SARS-CoV2 [60-62], and is rapidly becoming more widely used in society. The advanced oxidation process (AOP), which uses ozone plus an ozone catalyst, has previously been reported for wastewater from water and wastewater treatment plants [63-65]. In cases such as hospital wastewater, where it is virtually impossible to install large-scale treatment facilities or secure a new hospital site for wastewater treatment, the case for evaluating the effectiveness of ozone treatment for untreated raw wastewater, or in a broader sense, advanced wastewater treatment, is a new challenge of great social interest in the aspect of hospitals and the environment [66,67]. The effect of applying advanced wastewater treatment systems to hospital wastewater to reduce the burden of AMR in the environment was determined by clarifying the inactivation effect on ARB, ARGs, and antimicrobials and analyzing the characteristics of microorganisms in wastewater before and after treatment using meta-genomic analysis.”

2)    The results claim to demonstrate that ozone treatment enables effective degradation of the antimicrobial molecules, however, the results demonstrate only a simple disinfection process rather than degradation.

The only degradation which obtained, is transformation of the parent compounds (PCs) create only transformation products (TPs) that are slightly differ (chemically) from the parent compounds, however, these TPs mostly more persistent, less biodegradable and could be even more toxic than the PCs. The amount of ozone which uses in that study was enough for disinfection only and slightly transformed the PCs) – I can’t indicate on an AOP process and mechanism

Response:

As the reviewer pointed out, previous studies have reported that ozone treatment causes the decomposition of environmental chemicals and transformation products (TPs). As for these TPs, it has been shown that their lower molecular weight generally makes them less toxic than the parent compound, but there have been reports of increased toxicity for some substances. As far as the authors perceive, the antimicrobials examined in this study do not contain components whose toxicity increases by ozone treatment compared to that before the treatment. Even when harmful TPs are generated, they presumably become harmless after a sufficient treatment time due to the strong oxidizing power of ozone. Due to technical limitations, bioassay or other ecotoxicity impact assessments were not conducted in this study, and we consider that this part of the study remains a challenge for future investigations. According to the suggestion, a discussion on TPs has been added to the revised manuscript along with additional references.

3)    No sufficient explanation regarding the AOP applications in the Introduction

Response:

As suggested by reviewer and in relation to the 1), a description on the AOP applications has been included in section Introduction’ in the revised manuscript with additional references.

4)    Please show UPLC gradient conditions, the column type? and additional analytical values for MS. what was the LOD? LOQ? validation process of the analytical method?

Response:

As suggested by reviewer, UPLC gradient conditions, column type, additional analytical values for MS and validation of LOD, LOQ has been added to the supplemental materials as Table S6 and Table S7, and additional notations have been included in the revised manuscript.

5)    No matrix effect was indicated due to the high amount of organics in the non-sufficient treated WW?

Response:

In relation to 4), As suggested by reviewer and in relation to the 1), description of the quantification was performed by subtracting the blank data from the corresponding data yielded by the spiked sample solutions to account for matrix effects and losses during sample extraction is included in the revised manuscript.

6)    Line 239 - It is wearied, how the water became clearer and the measured parameters didn't change?!!

Response:

The main pigment color in the influent samples was urobilin (M.W. 590.71) in feces and urine, we speculated that such small compounds of urobilin could be easily degraded by ozonation, as observed in Figure 4. Therefore, ozone treatment inactivated most ARB and antimicrobials but did not change BOD or COD, although the treated water became clear.     

7)    Figure 4 is a table not a figure.

Response:

Thank you for the careful observation. Regarding Figure 4, if only the values of the measurement results were summarized, it would be in the Table. On the other hand, the range and number of numbers displayed varies, and in order to facilitate the visualization of the results obtained in this study, the values are presented as an image together with colored bars. Since the Instructions for Authors for this Journal stipulate that the format of the Table should be in the form of editable format (numbers only), we want to show these results as a composite Figure, which was also allowed in the Instructions for Authors. To make this easier to read, the notation the top 12 most abundant ARGs with a combined display of specific numerical values and a composite display of colored bars are shown in Figure 4 has been added to the main text.

 The limitations showed by the authors and the describe above require a major revision of the system, experiment, results and discussion prior to publication

Response:

Thank you for your very useful suggestions and comments, which contribute to the accuracy and novelty of this paper. Based on the reviewers' comments, we have revised and improved the present manuscript. The authors hope that the revised manuscript will be suitable for publication in your Journal.

Reviewer 2 Report

This manuscript investigated the impacts of an advanced ozonation-based continuous flow wastewater treatment system, and described the variation and risk that AMR, ARB and antimicrobials caused. The work is practical and interesting, and the advanced and reliable technologies are used to analysis the difference between ozonation-based and UV-LED system groups. Nevertheless, there are still many questions need to be address and mistakes need to be correct. Especially, authors should prepare their paper in correct English language including grammar, tense, spelling, presentation and so on.

Special comments:

1.     Graphical Abstract is suggested which make this part more concise.

2.     Line547, Table information content is marked incorrectly. And the format for citing supporting information is best consistent with the main text.

3.     Accession numbers of all raw read sequences used in the manuscript should be provided in the Materials and Methods section rather than cited in the Results section.

4.     Line548, it's better to check if the webpage is correct.

5.     Line174, “PBS” here are the first appearance, and you should give the full name.

6.     Line 59, “it is important to consider the indirect effects of the propagation of antimicrobial resistance genes (ARGs) on E. coli, which are widely distributed in the environment, by encouraging the development of new AMR”. It was mentioned in the Results section that it was acting as the most ubiquitous ESBL producers in line 282, which suggested to be briefly explained in the introduction as it is one of the arguments leading to the conclusion. Also, This part is mentioned multiple times in the main text, indicating its importance to the conclusion. Line 172. “To determine the efficacy of ozone treatment in inactivating potential β-lactam-resistant bacteria”.

7.     For bacteria, identifying the Genus level often cannot determine pathogenicity. It is recommended to identify at the Species level and examine the varieties of pathogenic bacteria carrying ARG before and after identification. The experiment adopted an ozone-based hospital wastewater treatment system,and a UV-LED group was set up in the experiment. A large number of bacteria were observed to be deactivated through the results of BTB agar and CHROMagar ESBL. It is suggested that in the following section of the Results, further explanation can be given on the different types of antimicrobial resistance genes that exist in surviving pathogenic bacteria. Currently, the general public's broad focus is on ESKAPE in specific hospital environments. “hospital wastewater may serve as a reservoir for AMR”. It is recommended to expand the focus range for the identification results of AMR, thus assessing the actual situation of ARB and antimicrobials in hospital wastewater and their environmental risk”,which can make the explanation of this part better demonstrate “regional security and the safety of the local population”.

8.     Parallel experiments were conducted on UV and ozone in the article, and the results were as expected. It is suggested that further elaboration on the advantages and disadvantages of combining ozone and ultraviolet light can be discussed just in the Discussion section.

Author Response

Reviewer2

Comments and Suggestions for Authors

This manuscript investigated the impacts of an advanced ozonation-based continuous flow wastewater treatment system, and described the variation and risk that AMR, ARB and antimicrobials caused. The work is practical and interesting, and the advanced and reliable technologies are used to analysis the difference between ozonation-based and UV-LED system groups. Nevertheless, there are still many questions need to be address and mistakes need to be correct. Especially, authors should prepare their paper in correct English language including grammar, tense, spelling, presentation and so on.

Special comments:

  1. Graphical Abstract is suggested which make this part more concise.

Response:

Graphical Abstract is included in the revised manuscript as suggested.

  1. Line547, Table information content is marked incorrectly. And the format for citing supporting information is best consistent with the main text.

Response:

The order of supplementary Tables was shifted off by one, therefore, the order has been corrected.    

  1. Accession numbers of all raw read sequences used in the manuscript should be provided in the Materials and Methods section rather than cited in the Results section.

Response:

The description “All raw read sequence files are available from the DRA/SRA database (Table S2)” was moved to the Materials and Methods section.

  1. Line548, it's better to check if the webpage is correct.

Response:

All webpages are correct and active (accessed date: March 23, 2003).

  1. Line174, “PBS” here are the first appearance, and you should give the full name.

Response:

PBS was changed to full description “phosphate-buffered saline”.

  1. Line 59, “it is important to consider the indirect effects of the propagation of antimicrobial resistance genes (ARGs) on E. coli, which are widely distributed in the environment, by encouraging the development of new AMR”. It was mentioned in the Results section that it was acting as “the most ubiquitous ESBL producers” in line 282, which suggested to be briefly explained in the introduction as it is one of the arguments leading to the conclusion. Also, This part is mentioned multiple times in the main text, indicating its importance to the conclusion. Line 172. “To determine the efficacy of ozone treatment in inactivating potential β-lactam-resistant bacteria”.

Response:

Thank you for the careful observation and invaluable contribution. The problem of antimicrobial resistance is widespread, and the early detection and control of antimicrobial resistance in highly pathogenic microorganisms is a pressing issue in terms of the direct health impact on patients in hospitals and other healthcare facilities. On the other hand, in the case of antimicrobial resistance in the environment, the problem of which microorganisms to monitor and the basis for such monitoring is still a work in progress, even with modern science and technology, even on a global scale. The antimicrobial-resistant E. coli (ESBL) discussed in this paper are not directly lethal, but recent research shows that ESBL are causing infections not only in healthcare facilities but also in the community. E. coli has been identified as an indicator microorganism in water systems. It is a representative example in our current paper, in part because of concerns about its potential to create a public health problem of significant magnitude through genetic effects and uptake with other microorganisms.

  1. For bacteria, identifying the Genus level often cannot determine pathogenicity. It is recommended to identify at the Species level and examine the varieties of pathogenic bacteria carrying ARG before and after identification. The experiment adopted an ozone-based hospital wastewater treatment system,and a UV-LED group was set up in the experiment. A large number of bacteria were observed to be deactivated through the results of BTB agar and CHROMagar ESBL. It is suggested that in the following section of the Results, further explanation can be given on the different types of antimicrobial resistance genes that exist in surviving pathogenic bacteria. Currently, the general public's broad focus is on ESKAPE in specific hospital environments. “hospital wastewater may serve as a reservoir for AMR”. It is recommended to expand the focus range for the identification results of AMR, thus “assessing the actual situation of ARB and antimicrobials in hospital wastewater and their environmental risk”,which can make the explanation of this part better demonstrate “regional security and the safety of the local population”.

Response:

We partially agree with the reviewer’s comment regarding the characterization of bacteria at the species level in the metagenomic analysis.

We supposed that susceptibility to either Ozone or UV treatment could depend on the basic features of the organism (GENUS level), not on the species level, whether it is pathogenic or not.

In addition, we presented the Figure 3 for two major genus of intestinal bacteria (Bacteroides, Prevotella) and four major genera among ESKAPE (Enterococcus faecium, Staphylococcus aureus, Klebsiella pneumoniae, Acinetobacter baumannii, Pseudomonas aeruginosa, Enterobacter).

Based on the AMR issue, persistent of Acinetobacter and Pseudomonas could play a major role in AMR reservoirs, although ozone treatment might act efficiently.

A few remarks were included in the discussion section.

  1. Parallel experiments were conducted on UV and ozone in the article, and the results were as expected. It is suggested that further elaboration on the advantages and disadvantages of combining ozone and ultraviolet light can be discussed just in the Discussion section.

Response:

According to this suggestion and a similar suggestion made by Reviewer 1, description of discussion of further elaboration on the advantages and disadvantages of combining ozone and ultraviolet light was included in the Discussion section with additional references.

Reviewer 3 Report

As a biostatistician with limited knowledge of Microbiology and Urban Planning, may I suggest the following methods to improve the paper:

- The authors could try Repeated Measures analysis with each microorganism and report if there was any statistical difference. See here: https://statistics.laerd.com/spss-tutorials/two-way-repeated-measures-anova-using-spss-statistics.php

- On Page 7 why are there spikes on different days for different microorganisms?

- Do you want to compare Ozone and UV- LED treatments? If yes, then as well repeated measures or Mixed Models could be used.

- My other question is on the cost of this water treatment setup? Is it cheaper than existing methods of dealing with waste water? A few statements on that would help. A detailed cost benefit analysis is beyond the scope of this paper. 

- Could this system be implemented in developing countries, where nosocomial drug-resistance is a major issue. 

Author Response

Reviewer 1

Comments and Suggestions for Authors

A major revision id require:

1) The treatment system is wrong. No primary and secondary treatment were conducted and indicated. The AOP (advanced oxidation process), which based on ozone or UV, always applied after secondary treatment. Therefore, you cannot call the system that was applied in that manuscript - an advanced treatment! You cannot apply AOP on raw WW and call it advanced treatment. The only treatment which was demonstrated in that manuscript – is a simple disinfection process that was done by a strong oxidant such as ozone and UV photons. For evidence, the oxidant was strong enough to disinfect bacteria but not strong enough to mineralize organics (BOD, COD, SS- didn’t change during the treatment!!!). Additionally, the high concentrations of the BOD, COD and SS, that characterize raw WW or insufficient treated effluent, may act as a scavengers in a real AOP treatment process (consumed immediately by the organics or and block (SS) the UV photons).

Response:

Thank you for the careful observation and invaluable contribution. Our research group understands the points raised by the reviewers about the processing method, and we discussed them before starting this project. First, regarding the advanced oxidation process (AOP), the hospital wastewater treatment investigation conducted in this study are totally different from those conducted at drinking water plants and wastewater treatment plants. The most unique feature of hospital wastewater treatment is that wastewater from existing hospital facilities is discharged directly into the public sewer system in Japan without treatment. This would mean that if a new hospital wastewater treatment system were to be installed, a large treatment facility would be required to occupy a certain amount of space in the hospital in the wastewater treatment area, or purchase a large new parcel of land next to the hospital and build a treatment facility. This trend is similar not only in Japan, but also in other parts of the world. Due to these practical limitations, it is difficult to install sedimentation tanks or biological treatment reactors, so this study is a new perspective and an unprecedented challenge to evaluate the effectiveness of measures that can be taken by hospital facilities that can directly treat raw sewage.

Response: I’m totally aware to HWW treatment worldwide, however, you cannot treat raw WW with advanced technologies due to particles which disturb the photons efficiency and the organic will scavenger the ozone . This is why a preliminary and secondary treatment is require, even in HWW! Still your study based only a simple disinfection process that was done by a strong oxidant such as ozone and UV photons. For evidence, the oxidant was strong enough to disinfect bacteria but not strong enough to mineralize organics (BOD, COD, SS- didn’t change during the treatment!!!). Additionally, the high concentrations of the BOD, COD and SS, that characterize raw WW or insufficient treated effluent, may act as a scavengers in a real AOP treatment process (consumed immediately by the organics or and block (SS) the UV photons).

Response from Authors:

Thank you very much for invaluable contribution for the novelty and accuracy of the present paper. We agree with the reviewer. Based on the findings in this review that the raw effluent treatment approach did not result in sufficient ozone and UV being fully effective, the terms AOP and advanced treatment are revised as disinfection process throughout the manuscript, and the main text before and after is also revised accordingly.

For this reason, the term "advanced treatment" is used in the text to refer to the broad term advanced wastewater treatment, which is also used for nitrogen and phosphorus removal in biological treatment before secondary treatment, rather than limiting the term directly to AOP as a specific form of treatment. There are parts of the content that are unclear and difficult to understand, so in accordance with the reviewers' opinions, we have revised the main text along with additional references to clarify that advanced treatment refers to advanced wastewater treatment using ozone treatment as follows:

“As part of the efforts to implement the hospital wastewater treatment system in society, a pioneering trial has begun to verify the effectiveness of measures utilized to reduce the environmental burden of hospital wastewater treatment system by conducting continuous treatment of the entire hospital wastewater before discharging it into the public sewer system. Therefore, in the present study, a continuous treatment system that can effectively treat hospital wastewater without any interference with hospital operations was developed. To achieve a more effective treatment, we additionally tested an ultraviolet light-emitting diode (UV-LED) [58,59], which has recently been shown to be effective in disinfecting pathogenic microorganisms, including SARS-CoV2 [60-62], and is rapidly becoming more widely used in society. The advanced oxidation process (AOP), which uses ozone plus an ozone catalyst, has previously been reported for wastewater from water and wastewater treatment plants [63-65]. In cases such as hospital wastewater, where it is virtually impossible to install large-scale treatment facilities or secure a new hospital site for wastewater treatment, the case for evaluating the effectiveness of ozone treatment for untreated raw wastewater, or in a broader sense, advanced wastewater treatment, is a new challenge of great social interest in the aspect of hospitals and the environment [66,67]. The effect of applying advanced wastewater treatment systems to hospital wastewater to reduce the burden of AMR in the environment was determined by clarifying the inactivation effect on ARB, ARGs, and antimicrobials and analyzing the characteristics of microorganisms in wastewater before and after treatment using meta-genomic analysis.”

Response: It's being improved but not enough. The claim that HWW cannot undergo preliminary and secondary treatment is not true. Again, you cannot skip these important steps and to use only advanced methods to degrade organics – I already explained why it is impossible and even the result of the study didn’t support that claim. Changing the words will not change the fact that UV and ozone were implemented on raw WW.

Response from Authors:

Thank you very much for invaluable contribution. As we mentioned in previous response, the terms AOP and advanced treatment are revised as disinfection process throughout the manuscript, and the main text before and after is also revised accordingly.

2) The results claim to demonstrate that ozone treatment enables effective degradation of the antimicrobial molecules, however, the results demonstrate only a simple disinfection process rather than degradation.

The only degradation which obtained, is transformation of the parent compounds (PCs) create only transformation products (TPs) that are slightly differ (chemically) from the parent compounds, however, these TPs mostly more persistent, less biodegradable and could be even more toxic than the PCs. The amount of ozone which uses in that study was enough for disinfection only and slightly transformed the PCs) – I can’t indicate on an AOP process and mechanism

Response:

As the reviewer pointed out, previous studies have reported that ozone treatment causes the decomposition of environmental chemicals and transformation products (TPs). As for these TPs, it has been shown that their lower molecular weight generally makes them less toxic than the parent compound, but there have been reports of increased toxicity for some substances. As far as the authors perceive, the antimicrobials examined in this study do not contain components whose toxicity increases by ozone treatment compared to that before the treatment. Even when harmful TPs are generated, they presumably become harmless after a sufficient treatment time due to the strong oxidizing power of ozone. Due to technical limitations, bioassay or other ecotoxicity impact assessments were not conducted in this study, and we consider that this part of the study remains a challenge for future investigations. According to the suggestion, a discussion on TPs has been added to the revised manuscript along with additional references.

Response: Did you check the claim “shown that their lower molecular weight generally makes them less toxic” this you check the obtained molecular weight of the obtained TPs? Did you check their toxicity? Most of the TPs are not differ by their MW from their parents and the toxicity could even increase! (Zilberman et al. 2023. Pharmaceuticals Transformation Products Formed by Ozonation - Is Degradation Occur? Molecules, 28, 1227)

Response from Authors:

Thank you for the careful observation and invaluable contribution. We agree with the reviewer on this point as well. According to the suggestion, we have included in the main text, citing the papers cited by the reviewers, that previous studies have shown that wastewater treatment produces substances that are more toxic than those before treatment. We also supplemented the discussion with references to the latest findings and perspectives on the evaluation of toxic effects of wastewater as follows:

“Previous studies have reported that ozone and/or UV treatment reduces ecotoxicological effects to approximately 1/10−1/20,000 compared to untreated compounds [47,98-100]. On the other hand, some researchers pointed the toxicity increase approximately 2−100 fold in some cases [99,100]. It is known that differences in susceptibility to these toxic effects occur in different target species, and that under conditions where multiple compounds coexist, weakening or strengthening effects would occur compared to exposure to a single substance [101,102]. Whole effluent toxicity (WET), as recommended by the US Environmental Protection Agency (EPA), which evaluates the toxic effects of a target water body as a whole using species at different trophic levels, would be a comprehensive approach to address these issues [103-105]. In addition, the strong oxidizing action of ozone or hydroxyl radicals can decrease the formation of transformation products by providing sufficient processing time and by acting in combination with catalysts, such as UV and hydrogen peroxide [46,106,107]. The results demonstrated that antimicrobials with an environmental impact were effectively removed before flowing into the environment and kept at a low level are notable as an effective measure to reduce the environmental impact caused by AMR.”

3) No sufficient explanation regarding the AOP applications in the Introduction

Response:

As suggested by reviewer and in relation to the 1), a description on the AOP applications has been included in section Introduction’ in the revised manuscript with additional references.

Response: OK

4) Please show UPLC gradient conditions, the column type? and additional analytical values for MS. what was the LOD? LOQ? validation process of the analytical method?

Response:

As suggested by reviewer, UPLC gradient conditions, column type, additional analytical values for MS and validation of LOD, LOQ has been added to the supplemental materials as Table S6 and Table S7, and additional notations have been included in the revised manuscript.

Response: OK

5) No matrix effect was indicated due to the high amount of organics in the non-sufficient treated WW?

Response:

In relation to 4), As suggested by reviewer and in relation to the 1), description of the quantification was performed by subtracting the blank data from the corresponding data yielded by the spiked sample solutions to account for matrix effects and losses during sample extraction is included in the revised manuscript.

Response: OK

6) Line 239 - It is wearied, how the water became clearer and the measured parameters didn't change?!!

Response:

The main pigment color in the influent samples was urobilin (M.W. 590.71) in feces and urine, we speculated that such small compounds of urobilin could be easily degraded by ozonation, as observed in Figure 4. Therefore, ozone treatment inactivated most ARB and antimicrobials but did not change BOD or COD, although the treated water became clear.

Response: What about the organic mixture in the raw WW? These will create turbidity, not the urobilin?! What about this claim? “Therefore, ozone treatment inactivated most ARB and antimicrobials but did not change BOD or COD, although the treated water became clear”

Response from Authors:

Thank you for the careful observation and invaluable contribution. What this part of the sentence means is that it has become clear that this treatment facility is capable of disinfecting ARB and antimicrobials, which are considered to be of immediate concern, although water quality items were not improved. We have revised the description in the main text to ''For the hospital wastewater treatment in this study, the treatment system was found to be capable of inactivating ARB and antimicrobials, which have become new environmental pollutants of concern in recent years, although some improvements, including water quality, have not yet been achieved.'' to make the meaning clearer.

Again, thank you very much for your very useful suggestions and comments, which contribute to the accuracy and novelty of this paper. Based on the reviewers' comments, we have revised and improved the present manuscript. We hope that the present manuscript now will be suitable for publication in your Journal.

7) Figure 4 is a table not a figure.

Response:

Thank you for the careful observation. Regarding Figure 4, if only the values of the measurement results were summarized, it would be in the Table. On the other hand, the range and number of numbers displayed varies, and in order to facilitate the visualization of the results obtained in this study, the values are presented as an image together with colored bars. Since the Instructions for Authors for this Journal stipulate that the format of the Table should be in the form of editable format (numbers only), we want to show these results as a composite Figure, which was also allowed in the Instructions for Authors. To make this easier to read, the notation the top 12 most abundant ARGs with a combined display of specific numerical values and a composite display of colored bars are shown in Figure 4 has been added to the main text.

The limitations showed by the authors and the describe above require a major revision of the system, experiment, results and discussion prior to publication

Response: OK

Response from Authors:

Thank you for your very useful suggestions and comments, which contribute to the accuracy and novelty of this paper. Based on the reviewers' comments, we have revised and improved the present manuscript. The authors hope that the revised manuscript will be suitable for publication in your Journal.

Round 2

Reviewer 1 Report

A major revision id require:

1) The treatment system is wrong. No primary and secondary treatment were conducted and indicated. The AOP (advanced oxidation process), which based on ozone or UV, always applied after secondary treatment. Therefore, you cannot call the system that was applied in that manuscript - an advanced treatment! You cannot apply AOP on raw WW and call it advanced treatment. The only treatment which was demonstrated in that manuscript – is a simple disinfection process that was done by a strong oxidant such as ozone and UV photons. For evidence, the oxidant was strong enough to disinfect bacteria but not strong enough to mineralize organics (BOD, COD, SS- didn’t change during the treatment!!!). Additionally, the high concentrations of the BOD, COD and SS, that characterize raw WW or insufficient treated effluent, may act as a scavengers in a real AOP treatment process (consumed immediately by the organics or and block (SS) the UV photons).

Response:

Thank you for the careful observation and invaluable contribution. Our research group understands the points raised by the reviewers about the processing method, and we discussed them before starting this project. First, regarding the advanced oxidation process (AOP), the hospital wastewater treatment investigation conducted in this study are totally different from those conducted at drinking water plants and wastewater treatment plants. The most unique feature of hospital wastewater treatment is that wastewater from existing hospital facilities is discharged directly into the public sewer system in Japan without treatment. This would mean that if a new hospital wastewater treatment system were to be installed, a large treatment facility would be required to occupy a certain amount of space in the hospital in the wastewater treatment area, or purchase a large new parcel of land next to the hospital and build a treatment facility. This trend is similar not only in Japan, but also in other parts of the world. Due to these practical limitations, it is difficult to install sedimentation tanks or biological treatment reactors, so this study is a new perspective and an unprecedented challenge to evaluate the effectiveness of measures that can be taken by hospital facilities that can directly treat raw sewage.

Response: I’m totally aware to HWW treatment worldwide, however, you cannot treat raw WW with advanced technologies due to particles which disturb the photons efficiency and the organic will scavenger the ozone . This is why a preliminary and secondary treatment is require, even in HWW! Still your study based only a simple disinfection process that was done by a strong oxidant such as ozone and UV photons. For evidence, the oxidant was strong enough to disinfect bacteria but not strong enough to mineralize organics (BOD, COD, SS- didn’t change during the treatment!!!). Additionally, the high concentrations of the BOD, COD and SS, that characterize raw WW or insufficient treated effluent, may act as a scavengers in a real AOP treatment process (consumed immediately by the organics or and block (SS) the UV photons).

For this reason, the term "advanced treatment" is used in the text to refer to the broad term advanced wastewater treatment, which is also used for nitrogen and phosphorus removal in biological treatment before secondary treatment, rather than limiting the term directly to AOP as a specific form of treatment. There are parts of the content that are unclear and difficult to understand, so in accordance with the reviewers' opinions, we have revised the main text along with additional references to clarify that advanced treatment refers to advanced wastewater treatment using ozone treatment as follows:

“As part of the efforts to implement the hospital wastewater treatment system in society, a pioneering trial has begun to verify the effectiveness of measures utilized to reduce the environmental burden of hospital wastewater treatment system by conducting continuous treatment of the entire hospital wastewater before discharging it into the public sewer system. Therefore, in the present study, a continuous treatment system that can effectively treat hospital wastewater without any interference with hospital operations was developed. To achieve a more effective treatment, we additionally tested an ultraviolet light-emitting diode (UV-LED) [58,59], which has recently been shown to be effective in disinfecting pathogenic microorganisms, including SARS-CoV2 [60-62], and is rapidly becoming more widely used in society. The advanced oxidation process (AOP), which uses ozone plus an ozone catalyst, has previously been reported for wastewater from water and wastewater treatment plants [63-65]. In cases such as hospital wastewater, where it is virtually impossible to install large-scale treatment facilities or secure a new hospital site for wastewater treatment, the case for evaluating the effectiveness of ozone treatment for untreated raw wastewater, or in a broader sense, advanced wastewater treatment, is a new challenge of great social interest in the aspect of hospitals and the environment [66,67]. The effect of applying advanced wastewater treatment systems to hospital wastewater to reduce the burden of AMR in the environment was determined by clarifying the inactivation effect on ARB, ARGs, and antimicrobials and analyzing the characteristics of microorganisms in wastewater before and after treatment using meta-genomic analysis.”

Response: It's being improved but not enough. The claim that HWW cannot undergo preliminary and secondary treatment is not true. Again, you cannot skip these important steps and to use only advanced methods to degrade organics – I already explained why it is impossible and even the result of the study didn’t support that claim. Changing the words will not change the fact that UV and ozone were implemented on raw WW.

2) The results claim to demonstrate that ozone treatment enables effective degradation of the antimicrobial molecules, however, the results demonstrate only a simple disinfection process rather than degradation.

The only degradation which obtained, is transformation of the parent compounds (PCs) create only transformation products (TPs) that are slightly differ (chemically) from the parent compounds, however, these TPs mostly more persistent, less biodegradable and could be even more toxic than the PCs. The amount of ozone which uses in that study was enough for disinfection only and slightly transformed the PCs) – I can’t indicate on an AOP process and mechanism

Response:

As the reviewer pointed out, previous studies have reported that ozone treatment causes the decomposition of environmental chemicals and transformation products (TPs). As for these TPs, it has been shown that their lower molecular weight generally makes them less toxic than the parent compound, but there have been reports of increased toxicity for some substances. As far as the authors perceive, the antimicrobials examined in this study do not contain components whose toxicity increases by ozone treatment compared to that before the treatment. Even when harmful TPs are generated, they presumably become harmless after a sufficient treatment time due to the strong oxidizing power of ozone. Due to technical limitations, bioassay or other ecotoxicity impact assessments were not conducted in this study, and we consider that this part of the study remains a challenge for future investigations. According to the suggestion, a discussion on TPs has been added to the revised manuscript along with additional references.

Response: Did you check the claim “shown that their lower molecular weight generally makes them less toxic” this you check the obtained molecular weight of the obtained TPs? Did you check their toxicity? Most of the TPs are not differ by their MW from their parents and the toxicity could even increase! (Zilberman et al. 2023. Pharmaceuticals Transformation Products Formed by Ozonation - Is Degradation Occur?  Molecules, 28, 1227)

3) No sufficient explanation regarding the AOP applications in the Introduction

Response:

As suggested by reviewer and in relation to the 1), a description on the AOP applications has been included in section Introduction’ in the revised manuscript with additional references.

Response: OK

4) Please show UPLC gradient conditions, the column type? and additional analytical values for MS. what was the LOD? LOQ? validation process of the analytical method?

Response:

As suggested by reviewer, UPLC gradient conditions, column type, additional analytical values for MS and validation of LOD, LOQ has been added to the supplemental materials as Table S6 and Table S7, and additional notations have been included in the revised manuscript.

Response: OK

5) No matrix effect was indicated due to the high amount of organics in the non-sufficient treated WW?

Response:

In relation to 4), As suggested by reviewer and in relation to the 1), description of the quantification was performed by subtracting the blank data from the corresponding data yielded by the spiked sample solutions to account for matrix effects and losses during sample extraction is included in the revised manuscript.

Response: OK

6) Line 239 - It is wearied, how the water became clearer and the measured parameters didn't change?!!

Response:

The main pigment color in the influent samples was urobilin (M.W. 590.71) in feces and urine, we speculated that such small compounds of urobilin could be easily degraded by ozonation, as observed in Figure 4. Therefore, ozone treatment inactivated most ARB and antimicrobials but did not change BOD or COD, although the treated water became clear.

Response: What about the organic mixture in the raw WW? These will create turbidity, not the urobilin?! What about this claim? “Therefore, ozone treatment inactivated most ARB and antimicrobials but did not change BOD or COD, although the treated water became clear

7) Figure 4 is a table not a figure.

Response:

Thank you for the careful observation. Regarding Figure 4, if only the values of the measurement results were summarized, it would be in the Table. On the other hand, the range and number of numbers displayed varies, and in order to facilitate the visualization of the results obtained in this study, the values are presented as an image together with colored bars. Since the Instructions for Authors for this Journal stipulate that the format of the Table should be in the form of editable format (numbers only), we want to show these results as a composite Figure, which was also allowed in the Instructions for Authors. To make this easier to read, the notation the top 12 most abundant ARGs with a combined display of specific numerical values and a composite display of colored bars are shown in Figure 4 has been added to the main text.

The limitations showed by the authors and the describe above require a major revision of the system, experiment, results and discussion prior to publication

Response: OK

Response:

Thank you for your very useful suggestions and comments, which contribute to the accuracy and novelty of this paper. Based on the reviewers' comments, we have revised and improved the present manuscript. The authors hope that the revised manuscript will be suitable for publication in your Journal.

Author Response

(The authors gave the same response as above.)

Reviewer 2 Report

The authors have revised the manuscript according to the comments, and the current form can now be accepted.

Author Response

Thank you for your very useful suggestions and comments, which contribute to the accuracy and novelty of this paper.